# Highly Effective Frontal Stunning Procedure Using a Pneumatic Penetrating Captive Bolt in Water Buffaloes

**DOI:** 10.3390/ani13010177

**Published:** 2023-01-03

**Authors:** Valeria Molnar-Fernández, Lohendy Muñoz-Vargas, Juan José Romero-Zúñiga, Gustavo Araya-Rodríguez

**Affiliations:** 1Escuela de Medicina Veterinaria, Universidad Nacional, Heredia 304-3000, Costa Rica; 2Laboratorio de Salud Pública y Alimentos, PIET, Escuela de Medicina Veterinaria, Universidad Nacional, Heredia 304-3000, Costa Rica; 3Programa de Investigación en Medicina Poblacional, Escuela de Medicina Veterinaria, Universidad Nacional, Heredia 304-3000, Costa Rica; 4Servicio Nacional de Salud Animal (SENASA), Ministerio de Agricultura y Ganadería, Heredia 3-3006, Costa Rica

**Keywords:** stunning, welfare, slaughter, water buffaloes, penetrating pneumatic captive bolt

## Abstract

**Simple Summary:**

Water buffalo meat and milk consumption has increased in recent years. This species is difficult to stun due to its skull conformation, affecting proper and effective stunning using conventional equipment. The objective of this study was to evaluate the stunning procedure of water buffaloes, using pneumatic mechanical stunning equipment most used in the American continent, by assessing an adequate site for frontal stunning. The results showed that the anatomical site proposed by the authors had a 95% effectiveness at the first shot. This finding is highly important as it significantly improves the welfare of water buffaloes taken to slaughter, one of the most stressful stages of their lives, providing a prompt unconscious state, as in cattle.

**Abstract:**

An effective frontal stunning procedure in water buffaloes was assessed using a pneumatic penetrating captive bolt (PPCB) with high air pressure. The study contemplated two phases; first, 352 buffaloes and 168 post-mortem heads were evaluated to determine the most effective anatomical site for stunning. Then, the second phase (*n* = 182) was used to validate the stunning procedure at the discovered anatomical site in the first phase, which was located 8 cm dorsal above the middle of the forehead on an “X” formed between the eyes and the base of the contralateral horns, and 2 cm lateral, avoiding the midline, where the skull tended to narrow. A total of 95.1% of buffaloes received effective stunning at the first shot with evidence of the presence of collapse, absence of rhythmic breathing, and absence of ocular reflexes (corneal and palpebral). There were no differences in the stunning efficacy by sex, breed, or skull thickness. These findings demonstrated that stunning with a PPCB at pressures of 1379–1516.8 KPa (200–220 pounds per square inch (psi)) in the site reported here produces a highly effective stunning at the first shot in water buffaloes.

## 1. Introduction

In Costa Rica, the stunning procedure used in water buffaloes (*Bubalus bubalis*) is frequently achieved by a frontal gunshot (.38 or 9 mm) or by a penetrating captive bolt (PCB) in poll position (occipital), and less frequently using a PCB at frontal level. However, the latest method, usually performed during cattle stunning, is not recommended in buffaloes due to species-dependent anatomical characteristics such as more profound and extensive frontal sinuses within the frontal and parietal rostrodorsal bones, complete and highly developed middle interfrontal septum, greater hardness of bone plates, and greater thickness of the hide, which hinders adequate and effective stunning [1,2,3,4,5,6,7]. However, the occipital approach is not effective enough to achieve adequate stunning because when shooting in this position, there is a risk of sectioning the spinal cord or cerebellum, causing animals to become paralyzed without losing consciousness; therefore, it should be interpreted with caution [2,5,8,9]. For this reason, European Union countries prohibit slaughtering cattle (Bos) in this particular anatomical site [10]. Similarly, firearms lead to occupational hazards and economic losses from head seizures to lead residues [7,11,12].

This study aimed to evaluate the effectiveness of a frontal stunning procedure in water buffaloes using a pneumatic penetrating captive bolt (PPCB) at high air pressures (200–220 psi), and to validate this procedure through the assessment of unconsciousness signs and post-mortem brain damage.

## 2. Materials and Methods

### 2.1. Ethical Statement

This study was conducted following the Costa Rican National Animal Welfare Procedures for animal handling and humane slaughter [13,14]. The Animal Welfare Commission at the School of Veterinary Medicine, Universidad Nacional, Costa Rica (UNA-EMV-CBBA-ACUE-007-2020), approved this project, as well as the Productive Species Animal Welfare Program at the National Animal Health Services (SENASA) of Costa Rica (SENASA-PBA-004-2021).

### 2.2. Facilities and Stunning Equipment

The abattoir facility is in Alajuela, Central Region of Costa Rica. It is a meat exporting company that slaughters approximately 350 animals per day at a rate of 40 per hour, mainly beef or dairy cattle. In addition, approximately 90 to 100 buffaloes are processed monthly. The stunning was performed with an expelled 9 cm pneumatic penetrating captive bolt (PPCB) Jarvis, model USSS-1 (Jarvis Products Corporation, Middletown, CT, USA). It uses 1379–1516.8 KPa (200–220 pounds per square inch (psi)) controlled through a pressurization system and hydraulic system manometer. All animals were stunned by properly trained operators by the establishment.

### 2.3. Animal Selection

All water buffaloes received at the slaughter facilities over the course of eight months during the study were involved and evaluated. They came from various farms and livestock auctions distributed throughout the country. Variables such as sex, breed, weight, origin, and carcass number were recorded to provide traceability for each included animal. The animals’ age was not considered within the variables since the establishment does not record it. Animals for which restraint at the stunning box could not be assured due to their body or horn conformation were excluded from the study.

### 2.4. Study Design

This study included a total of 534 buffaloes; it was conducted over the course of eight months in 2020–2021 divided into two different phases. The first phase (months 1, 2, and 3) included 352 buffaloes. It was performed to evaluate the stunning procedure using a PPCB in the frontal area, initially referenced by [15] described as follows: the intersection of two imaginary lines that join the upper and lower edges of the contralateral horns, avoiding the midline. In addition, the effective stunning was evaluated based on the presence of unconsciousness signs as described in Section 2.5. Additionally, a subset of approximately 50% of post-mortem heads were taken to verify the bolt entry point and the thickness of the skull.

The second phase (months 4, 5, 6, 7, and 8) included 182 buffaloes, validated by the newly proposed frontal stunning site using a PPCB. Effective stunning was evaluated based on the presence of unconsciousness signs as described in Section 2.5 of this document. A subset of 75 post-mortem heads were examined to measure the bolt entry point according to the proposed stunning site (located 8 cm dorsal above the middle of the forehead on an “X” formed between the eyes and the base of the contralateral horns, and 2 cm lateral, avoiding the midline), the skull thickness, and the degree of tissue damage caused by the PPCB. These were evaluated according to Figure 1.

Prior to the beginning of this phase, the slaughterhouse performed preventive maintenance for the entire pneumatic stunning system.

### 2.5. Stunning Efficacy Assessment

An effective stunning procedure was defined by the presence of the following unconsciousness signs [16]: a. presence of animal collapse (observing complete collapse inside the stunning box), b. absence of rhythmic breathing (evaluated by placing the hands in the nostrils), and c. absence of ocular reflexes (corneal reflex and palpebral reflex; evaluated by touching the cornea and eyelids with the fingertips, respectively). The animal was re-stunned if effective stunning was not achieved at the first shot. If unconsciousness signs were achieved, it was considered effective. If the second shot did not achieve an effective stunning, a 9 mm caliber Sig Sauer P250 (Sig Sauer, NH, USA) firearm was used for frontal stunning. Once proper stunning was achieved, the following 60 s included the animal being removed from the stunning box and being reevaluated before the exsanguination process.

During both study phases, the following information for each stunned animal was recorded: the slaughter number assigned for the establishment, sex, weight, breed, evaluation of consciousness signs (rhythmic breathing, collapse, ocular reflex), number of PPCB stuns, use of a secondary method of stunning (firearm), and bleeding time.

As opposed to cattle, for which re-stunning is recommended to be applied in a different place from the first shot [17], in the present study, the second shot was applied at the same shot hole as the first one if it was in the area recommended by the authors. This was intended to harness the damage generated by the first shot to the frontal bone, easing the bolt penetration of the second shot to generate an effective stun.

### 2.6. Post-Mortem Head Evaluation

Post-mortem examinations were conducted on the hornless, skinned, chilled heads preserved at 5 degrees Celsius 48 h after slaughter. The skulls were longitudinally split at the midline level using a bone saw. The bolt entry point and the skull thickness were registered during the first study phase. In the second phase, the bolt entry point (dorsal and lateral according to the proposed stunning site), the skull thickness, and the degree of tissue damage caused by the PPCB were evaluated according to Figure 1.

The laterality of entry was defined as “avoiding the midline”, due to the anatomical characteristics of water buffaloes’ skulls. In the post-mortem head evaluation, we measured how far the shot entry was from the midline, measured in centimeters.

### 2.7. Statistical Analysis

Before analysis, the data were checked for outliers or missing information. Therefore, there was no need to make any data imputation. Likewise, the variables bolt entry point (dorsal and lateral) according to the proposed stunning site, the skull thickness, and the degree of tissue damage caused by the PPCB were transformed to discrete variables. Finally, the limits defining the categories for each variable were established according to the researchers’ criteria.

Descriptive statistics were performed to describe the animals studied by calculating frequencies and measures of central tendency and dispersion globally and for each phase. For the continuous variables, the normality of their distribution was determined through the Shapiro–Wilk test.

Stunning efficacy was calculated as the percentage of buffaloes with total signs of unconsciousness divided by the total submitted for each attempt with the bolt. In the first phase (*n*= 352), overall efficacy was determined for the entire period and each of the three months. In the second phase (*n* = 182), only the overall efficacy for the whole period was calculated.

In the first phase, the overall efficacy of stunning was compared according to breed and sex using the chi-square test and live weight using the Kruskal–Wallis test. In addition, in the second phase, the skull thickness in its categorical form was added to the previous variables. Additionally, the frequency of signs indicating the absence of unconsciousness, according to the number of stunning attempts per bolt (first or second), was calculated for both phases.

In the second phase, a sample of 75 heads were evaluated to determine the degree of damage produced by PPCB, as described in Figure 1. Stunning efficacy was calculated by the number of attempts, skull thickness, breed, and sex. A difference in percentages determined by a chi-square test was performed for each variable.

All calculations were performed using SAS software ver. 9.4 (SAS Institute, Inc., Cary, NC, USA). A significance level of 0.05 was used for all comparisons.

## 3. Results

A total of 534 water buffaloes were evaluated during the study; a total of 381 were males (71.4%). Three major breeds were slaughtered and distributed as follows: crossbreeds, 315 (59.0%); Mediterranean, 150 (28.1%); and Murrah, 69 (12.9%). The average weight was 405.2 kg, including a minimum weight of 187.0 kg and a maximum of 746.0 kg.

### 3.1. Phase One: Evaluation of the Frontal Stunning Procedure

#### 3.1.1. Assessment of Stunning Efficacy

This phase lasted three months. A total of 352 buffaloes received for slaughter were evaluated. In the first month, the entry point “intersection of two imaginary lines that join the upper and lower edges of the contralateral horns, avoiding the midline” was used according to [15]. A total of 143 buffaloes were evaluated (40.6%), obtaining effective stunning at the first shot in 72 buffaloes (50.3%), at the second shot in 37 (25.8%), and the use of firearms was required in 34 buffaloes (23.7%). To evaluate the penetrating bolt localization, a subset of 62 heads were analyzed. Of those, 26 heads (41.9%) depicted a narrowed skull thickness approximately 8 cm above the middle of the forehead on an “X” formed between the eyes and the base of the contralateral horns [18], avoiding the midline. This specific site was associated with highly effective stunning at first shot. The anatomical reference point of captive bolt entry previously described was determined as thicker in buffaloes and presented a difficult replication of the stunning procedure since most buffaloes were dehorned and showed breed-dependent anatomical variations [15].

For Month 2, 141 buffaloes were evaluated (40.1%). Effective stunning at the first shot was achieved in 93 buffaloes (66.0%). The second shot was necessary for 26 buffaloes (18.4%) and the secondary method of stunning was required in 22 buffaloes (15.6%). During this month, 67 heads were evaluated to determine the accuracy of this new shot entry. In 40 heads (61.5%), the shot was in a range of 7.1–9.0 cm dorsal to the recommended area for cattle.

For Month 3, the assessment of stunning in the newly proposed site was repeated (Figure 2). A total of 68 buffaloes were evaluated (19.3%). Effective stunning at the first shot was achieved in 58 buffaloes (85.0%) The second shot was necessary in 6 buffaloes (9.0%) and in 4 buffaloes (6.0%), the use of a firearm was required. Thirty-nine heads were evaluated; in 35 heads (89.7%), the shot entry was observed to be between 7.1–9.0 cm dorsal to the recommended area for cattle. This showed a considerable improvement in both adequate first shot stunning and shot accuracy in the proposed area (Figure 3).

#### 3.1.2. Assessment of Stunning Efficacy by Breed, Sex, and Weight

In the first phase, crossbreeds were the most frequently received breed for slaughter and had the most remarkable stunning effectiveness at the first shot (Table 1). Furthermore, there were no differences between males and females, and the average weight at the first shot was 394 kg (range of 199–686 kg), 424.0 kg (range of 187–746 kg) at the second shot, and 463 kg (range of 243 kg–714 kg) when a firearm was necessary. Therefore, the weight of the animals did not influence the stunning efficacy.

#### 3.1.3. Assessment of Stunning Based on the Presence of Consciousness Signs

The total number of re-stunned buffaloes was 129, where 121 (34.4%) did not show collapse, 127 (36.1%) maintained rhythmic breathing, and 119 buffaloes (33.8%) maintained ocular reflexes (corneal reflex and palpebral reflex). The presence of rhythmic breathing was the most common consciousness sign for those animals.

A total of 61 buffaloes that did not receive effective stunning at the second shot required a gunshot, 41 (31.8%) had no collapse, 60 buffaloes (46.5%) maintained rhythmic breathing, and 39 buffaloes (30.2%) presented ocular reflexes (corneal reflex and palpebral reflex). The presence of rhythmic breathing was the most common consciousness sign for those animals.

### 3.2. Phase Two: Validation of the Proposed Stunning Site

#### 3.2.1. Assessment of Stunning Efficacy

Effective stunning at the first shot was achieved in 173 buffaloes (95.1%). In seven buffaloes, a second shot was required (3.9%), and only two buffaloes required the use of a firearm (1.1%). These results demonstrated that the second phase considerably increased the effectiveness of stunning at the first shot compared with the first phase (Figure 4).

#### 3.2.2. Evaluation of Stunning According to Sex, Breed, Weight, and Skull Thickness

Of the 173 buffaloes stunned at the first shot, 138 were males (79.7%). Of the seven buffaloes that required a re-stun, six were males (85.7%). Of the two buffaloes for which the use of a firearm was required, one was male and one was female.

According to breed, stunning at the first shot was effective in 69 Mediterranean buffaloes (39.9%), 68 crossbreed buffaloes (39.3%), and 36 Murrah buffaloes (20.8%).

Buffaloes stunned at the first shot weighed 386.1 kg on average (range 258.0–655.0 kg); re-stunned buffaloes were 514.1 kg on average (range of 401.0–625.0 kg), and buffaloes requiring bullet were 479.0 kg on average (range 403.0–555.0 kg).

Three skull thickness categories were defined as shown in Table 2. The mean skull thickness at the first shot was 3.2 cm, ranging from 1.5 to 6 cm. At the first shot, 28 heads (40.6%) presented a skull thickness of 3.1 to 5.0 cm, 39 heads (56.5%) of 1.0–3.0 cm, and 2 heads (2.9%) of 5.1 to 7.0 cm. At the second shot, three heads (50.0%) presented a skull thickness of 3.1 to 5.0 cm, two heads (33.3%) of 1.0 to 3.0 cm, and one head (16.7%) ranged from 5.1 to 7.0 cm.

Brain damage degree was classified from 0 to 5, based on Figure 1. The most frequent grade of damage was grade 2 (57.3%), followed by grade 5 (26.7%). No head presented a grade of 0 or 1. Non-significant differences were observed in the stunning procedure according to skull thickness, breed, and sex, suggesting high effectiveness of the stunning site regardless of those variables.

#### 3.2.3. Assessment of Stunning Based on the Presence of Consciousness Signs

Of the 182 evaluated buffaloes, 9 required a second shot due to the presence of more than one consciousness sign, including 4 buffaloes with an absence of collapse, presence of rhythmic breathing, and eye reflexes, 3 with an absence of collapse and presence of rhythmic respiration, and 2 buffaloes with a presence of rhythmic respiration.

Of the nine re-stunned buffaloes, only two buffaloes failed to be effectively stunned at the second shot; therefore, a firearm shot was required. The presence of rhythmic breathing was the most common consciousness sign for those animals.

#### 3.2.4. Evaluation of Brain Damage Caused by the PPCB

Seventy-five heads were evaluated. Of those, 69 heads received effective stunning at the first shot and 6 heads received effective stunning at the second shot. For those buffaloes stunned at the first shot, all 69 heads showed damage to the cerebral hemisphere (grade 2) associated with bolt entry. In 19 (27.5%), the degree of damage was classified as grade 5, being the second most frequent degree of brain damage (Figure 5 and Figure 6). For effective stunning at the second shot, the most frequent degree of brain damage detected was grade 2 (five of six heads), while one head had a damage degree of grade 5. No head presented grade 0 or 1 according to Figure 1.

Regarding weight, there was no association between the weight of the animals and the degree of damage generated.

#### 3.2.5. Evaluation of Stunning Efficacy According to Dorsal and Lateral Entry Range

Regarding effective stunning at the first shot, 69 heads were evaluated, finding that the dorsal entry in 47 heads (68.1%) was in a range of 6.0 to 9.0 cm; in 18 heads (26.1%), it was less than 6.0 cm, and in 4 heads (5.8%), it ranged from 9.1 to 11.0 cm.

For those six buffaloes that required a second shot, two heads presented a bolt entry in a range of 7.1 to 9.0 cm, and three heads presented an entry in the range of 6.1 to 7.0 cm. In one head, the entry was less than 6.0 cm.

Regarding the laterality of entry, at the first shot, seven heads (10.1%) had a bolt entry in a range of 0.0 to 1.0 cm. In 48 heads (69.7%), the entry was 1.1 to 3.0 cm. Seven heads (10.1%) had an entry in a range of 3.1 to 4.0 cm, and seven heads’ entries were (10.1%) >4.1 cm. At the second shot, four heads (66.7%) had an entry in a range of 1.1 to 3.0 cm. One head (16.6%) had an entry in a range of 3.1 to 4.0 cm and one head (16.6%) had an entry of >4.1 cm.

## 4. Discussions

To the best of our knowledge, this is the first published study that evaluated a highly effective PPCB stunning procedure in water buffaloes using a frontal anatomical entry site and demonstrated the association of this site to brain damage.

The objective of an adequate stunning with a PPCB is to cause the interruption of neurological function and subsequent insensitivity so that it does not produce pain or stress in the animal and guarantees a state of unconsciousness before hoisting, skinning, or any other invasive procedure [2,4,6,16,18,19,20]. Adequate stunning induces a state of unconsciousness through significant damage through one or more of the following pathways: in the cerebral hemispheres, in thalamic structures (reticular formation), or at the level of the ascending reticular activation system (SARA) [21].

Brain damage caused by a penetrating captive bolt is achieved through two main effects: First, percussive damage produces a shock wave through the brain, generating pressure gradients that injure the tissue and produce disturbances in the blood flow. In addition, this percussive effect can generate a brain herniation at the level of the tentorium, which creates compression in the brain stem, producing a decrease in or cessation of breathing or the heartbeat. Second, the mechanical destruction produced by the bolt in its path compresses the brain tissue and blood vessels, generating a hemorrhage that affects the supply of oxygen and nutrients. Moreover, when the bolt retracts, it produces a vacuum effect that generates more significant damage to neuronal axons and blood vessels in brain tissue. This damage causes neuronal depolarization of the cerebral hemispheres and potentially the brainstem, which alters the normal functioning of brain neurons. One of the main side effects of PCB stunning is the damage caused by local fragmentation of bone tissue, displacing parts of bone, hair, and skin within the brain tissue increasing the physical damage [21,22]. This was evidenced in the post-mortem evaluation of the analyzed heads, in which damage was found at different brain tissue levels, from the bolt entry hemisphere to thalamus (Figure 6).

The presence of signs of unconsciousness of high discriminatory power evaluated in this study demonstrated alterations at the level of reticular formation. In addition, cessation of breathing induces or contributes to brain anoxia [5,8,23]. However, rhythmic breathing could be present in an unconscious animal, but with the cessation of rhythmic breathing, it is considered to be unconscious or dead [8]; even so, those unconscious animals could breathe if only the midbrain is affected, without affecting the caudal and rostral protrusion of the medulla [24].

At present, frontal stunning with firearms or occipital entry with a penetrating captive bolt is recommended to stun water buffaloes due to difficulties given by the conformation of their frontal bones [1,6,11,22,25] with risks associated with both techniques (occupational risks and animal welfare violations) [7]. In the first phase of the present study, the entry site described by [15] was changed, showing a considerable improvement in stunning effectiveness at the first shot, rising from 50.3% to 85.0% (Figure 3). This considerable difference was due to a slight narrowing in thickness of the skull in the proposed entry site of the present study, which provides an anatomical window that facilitates frontal stunning in this species.

The study’s second phase contemplated the repeatability and validation of the effective stunning process proposed in this study. The presence of all unconsciousness signs in 95.1% of animals in the first shot proved that the new stunning site is ideal for frontal stunning in water buffaloes presented for slaughter. This strengthens the findings from the evaluation of the post-mortem heads, where it was evidenced that all analyzed brains showed evident damage at the cerebral hemisphere, with deformation of the brain tissue by the groove of the bolt with abundant hemorrhages and bone fragmentations. This is consistent with brain damage described in other studies [17,23,26]. Severe bleeding or tissue damage to the brainstem is considered the most reliable indication of massive brain trauma [23,27]. In addition, the increased levels of hemorrhage generated by the PPCB reduce the blood supply to the brain tissues with the consequent lack of oxygen and nutrients, which alters the intra- and extracellular biochemical balance [18]. Likewise, additional damage produced by bone fragmentations at different depths of the buffalo brains was evidenced. These findings are in agreement with a study conducted in cattle where bone fragmentation and bone penetration after a bolt hit can act as secondary missiles, causing more brain tissue damage (Figure 6) [17].

Others have mentioned the need to investigate a longer length bolt, which could reach the thalamus, but this has the eventual problem of bolt retraction [7]. Some authors reported that conventional penetrating devices with a bolt length of 90 mm are barely acceptable for stunning buffaloes in the frontal position [22]; however, the findings in the present study demonstrated that a PPCB used at high air pressures provides an adequate state of unconsciousness and brain damage in water buffaloes and is even more important than the bolt length. Further, these damages are not achieved with manual devices as demonstrated in other studies [7,15,28]. In our study, damage was evidenced at the corpus callosum, thalamus, and at the base of the cranial vault (7.3%, 10.1%, and 27.5%, respectively) in the evaluated heads. Therefore, the use of the PPCB Jarvis USSS-1 at high air pressures of 200–220 Psi was efficient in causing damage in brain regions responsible for maintaining the state of consciousness in water buffaloes (Figure 5). These findings are similar those reported in Nelore cattle [18]. Other studies have also used this stunning device in cattle, finding that variations in air pressure were not the main determinant of stunning success, but instead the speed related to the resulting kinetic energy that was administered to the brain. Therefore, at higher air pressures, more extensive brain damage was generated [29]. Hence, the findings reported in our study demonstrated that the new frontal approach produces adequate desensitization but also large-scale involvement of brain tissue. In addition, these findings demonstrate that despite the distance between the outer surface of the head and the brain sites of interest being larger than the size of the standard bolt, the PPCB used here achieves effective stunning evidenced by the damage produced (the brain tissue damage by the bolt and the side effects of the stunner system) without showing problems in its retraction, as has been seen in other captive bolt devices in other studies. [5].

Before starting the second phase, the compressed air maintenance unit was changed as part of the preventive maintenance of the stunning equipment. This unit filters the air to keep it free of impurities by trapping contaminants such as water, dust, and oil. In addition, it constantly regulates the air pressure to ensure that the equipment receives the necessary supply and lubricates the compressed air to reduce friction between the machine’s moving parts. We consider that this change in this unit provided significant improvements in the effectiveness of stunning since it enhanced the damage caused by the PPCB at the point of entry discovered in this study. Additionally, it reinforced what has been mentioned in other studies that indicate that air supply is a crucial part and should be maintained in adequate conditions [23,30].

In the validation phase, seven buffaloes required a second shot (3.9%) to produce an effective stunning, and only two buffaloes required the use of a firearm (1.1%). Of these nine buffaloes, the presence of rhythmic breathing was evident, and it was the only consciousness sign manifested in three of them; however, ocular reflexes were not observed in these animals, which could suggest a state of shallow unconsciousness and not necessarily a state of consciousness of the animal. Therefore, the presence of rhythmic breathing with the absence of ocular reflexes (corneal and palpebral) may not constitute inefficient stunning, as previously mentioned [8,23]. However, more studies are needed to evaluate this important aspect in water buffaloes.

Similarly, some buffaloes manifested partial eye rotation, nystagmus, and cervical flexion, which are signs of unconsciousness of low discriminatory power according to [8]. In these cases, the four unconsciousness signs of high discriminatory power were monitored, and there was no evidenced return to consciousness. The presence of low discriminatory power might suggest a shallow level of unconsciousness; however, the bleeding time used in this study guaranteed a quick death that prevented the possibility of a return to consciousness. In these cases, the authors suggest that a second shot could be applied (also known as a “security stun”) provided that the four unconsciousness signs of high discriminatory power are present in the stunning evaluation. This concept of a security stun has been previously mentioned by [21,31] for cattle stunning, and has been reinforced by what is mentioned by others [18], indicating that the use of a second shot would probably cause a greater degree of damage to the brain and a massive hemorrhage, which supports the findings of the present study.

The bleeding time was less than 60 s, sectioning the large vessels (carotid and jugular) at the level of the chest entrance to generate a massive loss of blood volume to ensure a safe and rapid death of the animal before it could recover its consciousness. This cut was performed using a well-maintained and sharp knife to prevent the formation of occluding clots in the sectioned vessels. This is important because, in bovids, the vertebral artery remains intact during slaughter by supplying oxygenated blood to the brain [5,6,21].

The mean skull thickness evaluated at post-mortem for buffaloes stunned at the first shot was 3.2 cm (range of 1.5 to 6.0 cm); there were three animals with skull thicknesses between 5.0 to 7.0 cm that had an effective stunning at the first shot with evident damage to the cerebral hemisphere. This mean thickness is similar to that of the study conducted by [26], in which the skull thickness of the buffaloes studied in a crown shot approach were between 3 to 4 cm; nevertheless, all presented collapse, and one had an absence of rhythmic breathing and corneal reflex; however, there was a return of rhythmic breathing. There is a consistent difference in our results, since 95.1% presented collapse, absence of rhythmic breathing, and absence of ocular reflexes after stunning at the first shot. This difference in the effectiveness of stunning at the first shot could be explained by the different PPCB used at a specific frontal site compared to the captive bolt gun used by [5]. In addition, a statistical tendency was shown in that the animals requiring bullets were those with the largest values of skull thickness.

The accuracy of the shooter was evaluated in the post-mortem heads during the validation phase, and it was observed that most of the buffaloes were shot near the site suggested in this study, being approximately 2 cm above or below the dorsal point and 1 cm lateral or medial of the midline. This variation may be due to differences in skull conformation between different buffalo breeds and brain position. Even so, the stunning was effective due to two factors: the area proposed in this study directs the bolt pin towards the thalamic zone, and the equipment used was powerful enough to generate the expected damage, in addition to the fact that the buffaloes were restrained using a head holder, which assisted the operator in locating this newly validated entry point. As indicated by [2], head restrainers can improve the stunning performance by decreasing the time spent in the stunning box compared to animals stunned without the device. This reinforces the importance of training slaughter workers for adequate practices in animal handling and operational practices, including during the stunning of water buffaloes.

## 5. Conclusions

The study demonstrates that the use of a PPCB Jarvis USSS-1 with a pressure of 200 to 220 PSI positioned at the frontal level 8 cm above the middle of the forehead on an “X” formed between the eyes and the base of the contralateral horns, and 2 cm lateral avoiding the midline, perpendicular to the skull, produces effective stunning at the first shot in the water buffaloes. Furthermore, it generates considerable damage at the brain level and guarantees the presence of the main unconsciousness signs of high discriminatory power regardless of sex, breed, and weight.

The constant training and empathy of the personnel in charge of stunning the buffaloes are essential to achieve effective stunning. In addition, efforts should be made to achieve shooting accuracy as close as possible to the new benchmark found in this study.

This present study is a first in demonstrating effective stunning at the frontal level in water buffaloes, as we consider the variables described here to positively impact the well-being of this species by reducing the anguish, pain, and stress at the time of its sacrifice by avoiding the use of inadequate or disallowed entry sites. In addition, it decreases the use of firearms in processing rooms that put both personnel and animals at risk.

## Figures and Tables

**Figure 1 animals-13-00177-f001:**
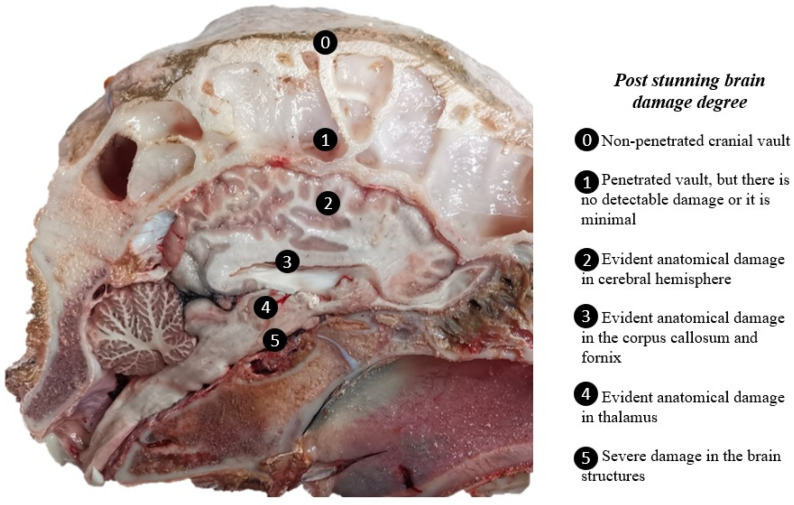
Midsagittal section of a buffalo head showing the post-stunning head and brain damage degree classification proposed in this study.

**Figure 2 animals-13-00177-f002:**
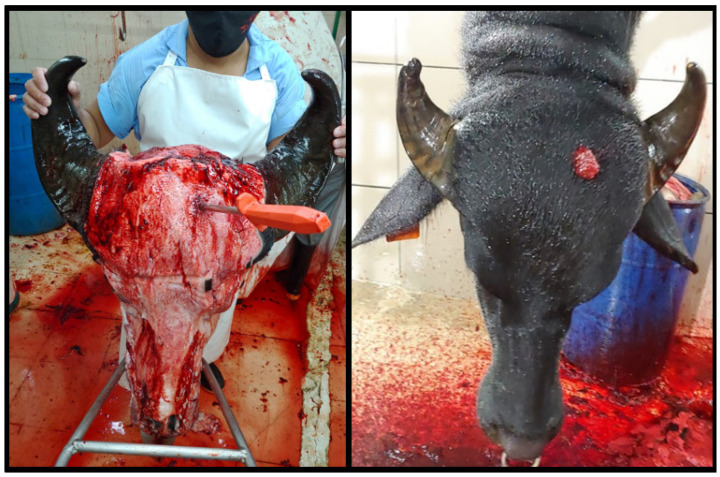
New proposed frontal stunning site (8 cm above the reference point of cattle and approximately 2 cm lateral avoiding midline) showed in three Mediterranean Buffaloes.

**Figure 3 animals-13-00177-f003:**
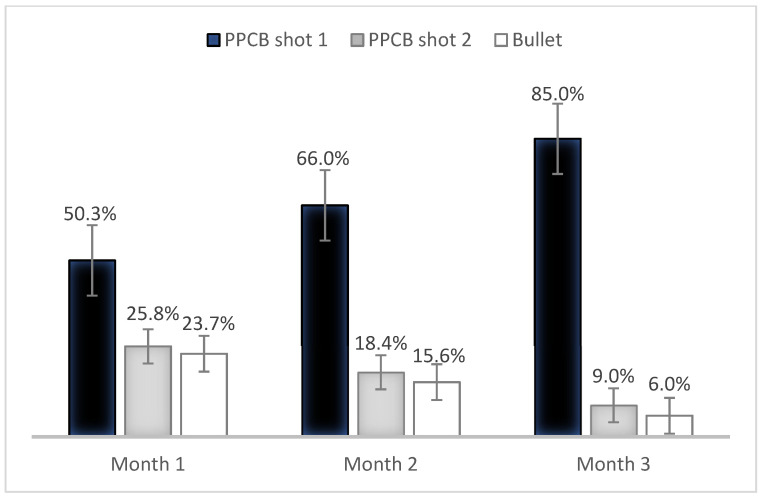
Frequencies of stunning effectiveness during the first phase of the study. Standard error bars are presented.

**Figure 4 animals-13-00177-f004:**
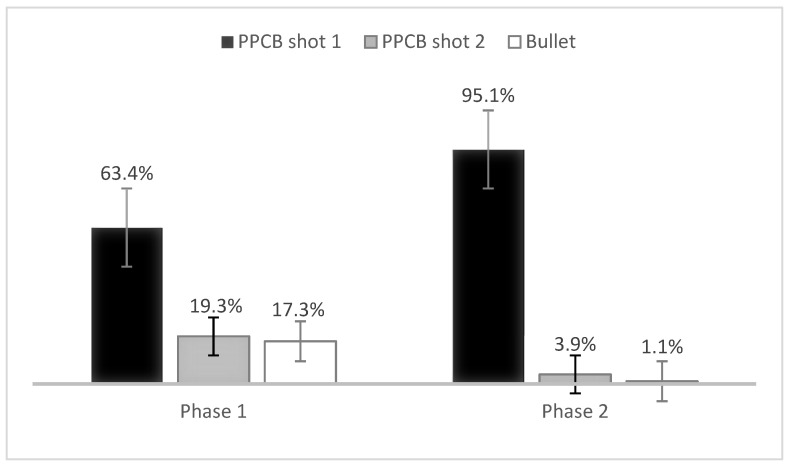
Stunning efficacy at first shot, second shot, and with firearm (bullet) for phases 1 and 2 of the study. Standard error bars are presented.

**Figure 5 animals-13-00177-f005:**
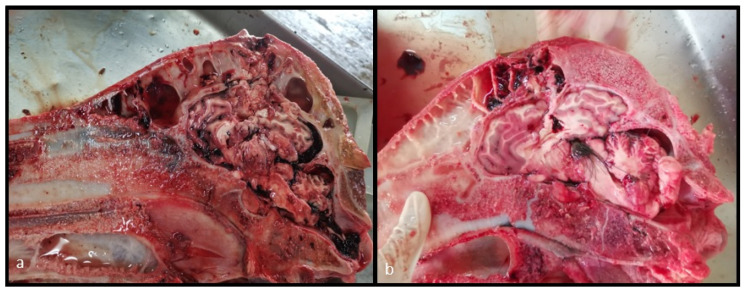
Heads of water buffaloes stunned at the first shot in the proposed area with their evident degree of damage at the level of the cerebral hemisphere. Massive destruction of brain tissue with abundant hemorrhage is observed (**a**); cerebral hemorrhage and malformation of fornix with evidence of hide segment in thalamus (**b**).

**Figure 6 animals-13-00177-f006:**
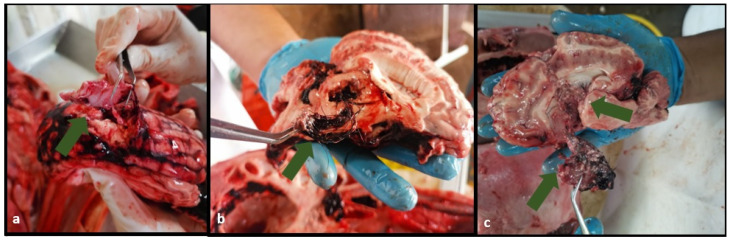
Water buffalo brain with abundant bleeding and extraction of a large bone fragment (green arrow) (**a**). Presence of hide and hair in the thalamus region due to the bolt’s impact on the left cerebral hemisphere (**b**). Left cerebral hemisphere sectioned by the bolt’s passage and bone fragment removal at the level of the thalamus (**c**).

**Table 1 animals-13-00177-t001:** Validation of anatomical site for an effective stunning procedure using a Pneumatic Penetrating Captive Bolt (PPCB) according to breed, sex, and weight in 534 buffaloes.

Phase	Stunning Shot	*n* (%)	Mean Kg (St. Dv)	Sex *n* (%)		Breed *n* (%)		
				Male	Female	Mixed	Med	Mur
**Phase 1**	PPCB shot 1	223 (63.4)	394.19 (77.5)	152 (64.4)	71 (61.2)	179 (73.4)	36 (46.1)	8 (26.7)
Glardon et al. (2018) [15] stunner position	PPCB shot 2	69 (19.6)	424.4 (103.6)	44 (18.6)	25 (21.6)	41 (16.8)	20 (25.6)	8 (26.7)
	Firearm	60 (17)	463.46 (93.9)	40 (17.0)	20 (17.2)	24 (9.8)	22 (28.2)	14(46.6)
	*n* (%)	352 (100)		236 (67.1)	116 (32.9)	244 (69.3)	78 (22.2)	30 (8.5)
**Phase 2**	PPCB shot 1	173 (95.1)	386.07 (65.1)	138 (95.2)	35 (94.6)	68 (95.7)	69 (95.8)	36(92.3)
New proposed stunner position	PPCB shot 2	7 (3.8)	514.14 (88.9)	6 (4.1)	1 (2.7)	2 (2.8)	2 (2.8)	3 (7.7)
	Firearm	2 (1.1)	479 (107.5)	1 (0.7)	1 (2.7)	1 (1.4)	1 (1.4)	0
	*n* (%)	182 (100)		145 (80)	37 (20)	71 (39)	72 (39.6)	39 (21.4)
***p*-value ***			<0.0001 ^#^	0.2605		0.02		

* A *p*-value ≤ 0.05 was considered as significant in all comparisons. ^#^ A Kruskal–Wallis test was used for weight comparisons between stunning methods.

**Table 2 animals-13-00177-t002:** Brain damage degree for an effective stunning procedure using a Pneumatic Penetrating Captive Bolt (PPCB) according to the number of shots, skull thickness, breed, and sex in 75 buffaloes during phase 2.

Variable	Brain Damage Degree, *n* (%)					*p*-Value *
**Effective Stunning Shot**	**Degree 2**	**Degree 3**	**Degree 4**	**Degree 5**	**Total**	<0.0001
First	38 (50.7)	5 (6.7)	7 (9.3)	19 (25.3)	69 (92.0)	
Second	5 (6.7)	0 (0.0)	0 (0.0)	1 (1.3)	6 (8.0)	
All	43 (57.3)	5 (6.7)	7 (9.3)	20 (26.7)	75 (100.0)	
**Skull thickness (cm)**						0.15
1.0–3.0	22 (29.3)	1 (1.3)	5 (6.7)	13 (17.3)	41 (54.7)	
3.1–5.0	18 (24.0)	4 (5.3)	2 (2.7)	7 (9.3)	31 (41.3)	
5.1–7.0	3 (4.0)	0 (0.0)	0 (0.0)	0 (0.0)	3 (4.0)	
**Breed**						0.6
C (crossbreed)	14 (18.7)	2 (2.7)	2 (2.7)	5 (6.7)	23 (30.7)	
ME (Mediterranean)	23 (30.7)	2 (2.7)	3 (4.0)	12 (16.0)	40 (53.3)	
MU (Murray)	6 (8.0)	1 (1.3)	2 (2.7)	3 (4.0)	12 (16.0)	
**Sex**						0.54
Female	9 (12.0)	0 (0.0)	1 (1.3)	2 (2.7)	12 (16.0)	
Male	34 (45.3)	5 (6.7)	6 (8.0)	18 (24.0)	63 (84.0)	

* A *p*-value ≤ 0.05 was considered as significant in all comparisons.

## Data Availability

The data presented in this study are available upon request from the corresponding author. The data are not publicly available due to the fact that the information is of a private and certificated export slaughterhouse.

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
