# Peer review of "Highly Effective Frontal Stunning Procedure Using a Pneumatic Penetrating Captive Bolt in Water Buffaloes"

_animals, 2023, doi:10.3390/ani13010177_

Round 1

Reviewer 1 Report

69: It suggest using S.I Metric units for a scientific paper.

98: Not a complete sentence?

125: It is not clear what is intended by this statement.

157: Ut is difficult to compare Phase 1 and Phase 2 as it appears that use of the new shot entry technique began in Month 2 and 3 in Phase 1.  Can it be made clear what differences in technique were employed as the changes in frequencies are not explained?

160: use footnote referencing  [28]

240: Give more specific description of how brain damage degree was evaluated.  Presumably the level of soft tissue disturbance in each of the anatomical region was evaluated in order to classify it in each degree category.

268: explain how "laterality of entry was defined and measured.

304: The results do not include any record of the specific damage described here from the post- mortems.

332: If this were evident in all cases this should be specified and documented in the Results.

337: Is this something claimed by this study?  What is inferred by "alters the intra- and extracellular biochemical balance"?

368: It seems very difficult to differentiate the effect of changing the compressed air maintenance unit and the effect of training the personnel on the changes between Phase 1 and 2.  There should be a better attempt to explain this.

408: Formation of what? Occluding clots?

Author Response

Response to Reviewer 1 Comments

69: It suggest using S.I Metric units for a scientific paper.

Response 69: Yes, the correct S.I. metric units are “1379 – 1516.8 KPa” (200-220 Psi). We´ll include this data in the paper as an appropriate reference. However, it is important to clarify the Psi units are used by the slaughterhouse to control air supply to the PPCB and we would like to use it because its practical use on ongoing monitoring.

98: Not a complete sentence?

Response 98: Agree. It´ll be changed to “Prior to the beginning of this phase, the slaughterhouse performed a preventive maintenance in the entire hydraulic stunning system.”

125: It is not clear what is intended by this statement.

Response 125: The intention of this statement was to clarify that the data obtained in this investigation did not require any data imputation.

157: Ut is difficult to compare Phase 1 and Phase 2 as it appears that use of the new shot entry technique began in Month 2 and 3 in Phase 1.  Can it be made clear what differences in technique were employed as the changes in frequencies are not explained?

Response 157: It is important to clarify that Phase 2 is a consecutive step of Phase 1. Phase 2 uses the same anatomic location as used in Phase 1. During Phase 2 two major determinants were improved: the stunning personnel was in a learning curve process to adapt a newer site to improve the accuracy of the buffaloes stunning procedure. The increased values regarding effective stunning at the first shot obtained in Month 2 and specially in Month 3 (85%) lead us to the conclusion that this was an appropriate frontal approach.

After a planned preventive maintenance of the PPCB, and the change the compressed air maintenance unit, there was a considerable improvement in the stunning results. In addition, the stunning personnel were better trained to identify the new shot entry. Those situations, led as to define both Phases.

Phase 1 evaluated the feasibility of frontal stunning with Jarvis USSS-1 PPCB, and in Phase 2 validated the effectiveness and repeatability of adequate stunning in this proposed area.

160: use footnote referencing [28]

Response 160: Agree. We´re going to modify as request.

240: Give more specific description of how brain damage degree was evaluated.  Presumably the level of soft tissue disturbance in each of the anatomical region was evaluated in order to classify it in each degree category.

Response 240: The brain damage degree was evaluated as described in Figure 1.

268: explain how "laterality of entry was defined and measured.

Response 268: The laterality of entry was defined as “avoiding the midline” (line 166), due to the anatomical characteristics of water buffalo’s skull. In the postmortem heads evaluation, we measured how far the shot entry was from the midline. We believe it is appropriate to include this information in the methodology 2.6. Postmortem heads evaluation

304: The results do not include any record of the specific damage described here from the post- mortems.

Response 304: They are shown in the Table 2 in the Results section.

332: If this were evident in all cases this should be specified and documented in the Results.

Response 332: They are presented in the Table 2 in the Results section. According to Figure 1, the brain damage classification is depicted. Table 2 the findings of the post mortem heads evaluated are shown. All buffaloes heads shown a grade 2 level damage, at least.

337: Is this something claimed by this study?  What is inferred by "alters the intra- and extracellular biochemical balance"?

Response 337: It is not claimed by our study. We consider the level of hemorrhage and structural brain damage produced by the Jarvis USSS-1 used in our study, generates physiological alterations in water buffaloes as in cattle in reference 18.

368: It seems very difficult to differentiate the effect of changing the compressed air maintenance unit and the effect of training the personnel on the changes between Phase 1 and 2.  There should be a better attempt to explain this.

Response 368: Please refer to the answer we provide on the response of 157.

408: Formation of what? Occluding clots?

Response 408: Yes, the correct sentence is “This cut was performed using a well-maintained and sharp knife to prevent the formation of occluding clots in the sectioned vessels.”, it´ll be changed.

Reviewer 2 Report

1) The main question addressed by the research is how to improve animal protection during slaughter.

2) The topic is field-relevant, because the authors address stunning, which is critical in reducing animal suffering; it is original, since a more performative and specie-specific tool is proposed and valued.  

3) The proposal is new in the filed because it focuses on water buffaloes rather than cattle.  

4) The methodology is nicely structured. Anyway, the authors could consider improving the description of the method and the requirements needed for the personnel.  

5) The conclusions are consistent with the evidence and arguments presented and they address the main question posed, but an additional element (the empathy of the personnel in charge of stunning the buffaloes) was added that should have been discussed beforehand and contextualised better.  

6) The references are appropriate  

7) Tables and figures are appropriate. Perhaps some more signs can be added to aid their comprehension.

Lines 98–99: The verb seems missing from the sentence.

Lines 107–108: Has the cause of the failure of the primary method of stunning been investigated?

Lines 378–382: This information would be suitable in the chapter Methods, since it is part of the description of the method.

Line 408: The sentence seems incomplete. 

Line 445: Please explain the role of empathy of the personnel in charge of stunning and whether its level was checked and evaluated, as it is never cited in the paper. 

Author Response

Response to Reviewer 2 Comments

Lines 98–99: The verb seems missing from the sentence.

Response 98-99: Authors appreciate this observation. It´ll be changed to “Prior to the beginning of this phase, the slaughterhouse performed a preventive maintenance in the entire hydraulic stunning system.”

Lines 107–108: Has the cause of the failure of the primary method of stunning been investigated?

Response 107-108: Yes, it has been investigated. Knowing the difficulty of frontal stunning in water buffaloes due to their skull conformation. the stunning personnel was in a learning curve process to adapt a newer site to improve the accuracy of the buffaloes stunning procedure. The increased values regarding effective stunning at the first shot obtained in Month 2 and specially in Month 3 (85%) lead us to the conclusion that this was an appropriate frontal approach.

After a planned preventive maintenance of the PPCB, and the change the compressed air maintenance unit, there was a considerable improvement in the stunning results. In addition, the stunning personnel were better trained to identify the new shot entry. Those situations, led as to define both Phases.

Lines 378–382: This information would be suitable in the chapter Methods, since it is part of the description of the method.

Response 378-382: Authors agree. We will change this paragraph and will be introduced in the point 2.5 of Materials and Methods according to the reviewer recommendation.

Line 408: The sentence seems incomplete. 

 Response 408: Yes, the correct sentence is “This cut was performed using a well-maintained and sharp knife to prevent the formation of occluding clots in the sectioned vessels.”

Line 445: Please explain the role of empathy of the personnel in charge of stunning and whether its level was checked and evaluated, as it is never cited in the paper. 

Response 445: The evaluation of the empathy of the personnel in charge of the stunning was considered by the authors based on the demonstration of compassion and respect towards the animal at the most critical moment (the sacrifice) in this process. The adequate disposition to learn, improve and execute that was taught for this study was crucial, not only for the stunning personnel but also for the animal welfare policy implemented in the slaughterhouse, seeking to generate a culture of respect for animals and facility the environment. From a quantitative point of view, the evaluation was carried out based on the number of shots fired, the accuracy of the hit, and the evaluation of the signs of unconsciousness that they carried out together with the authors during the on-site evaluations.

Reviewer 3 Report

The manuscript titled by Molnar-Fernandez et al. ''Highly effective frontal stunning procedure using a pneumatic penetrating captive bolt in water buffaloes''  deal with the effectiveness of stunning using pneumatic penetrating captive bolt in water buffaloes in different penetrating sites (3 sites-3 months during phase I and 1 new sits during phase II of study).

I suggest to take in the consideration the following comments:

the factor of animal’s age which is not considered in the study can be play important role on the stunning effectiveness

line 66: you indicated that the abattoir slaughters approximately 90 to 100 buffaloes are processed 66 monthly, but in the study at  phase I/ month 1 and 2 of study 143 and 141 buffaloes were slaughtered, respectively.

lines 80-88: in paragraph study design you indicated that during phase I of study one site of stunning was used '' in a frontal area, initially referenced by Glardon et al. (2018): the intersection of two imaginary lines that join the upper and lower edges of the contralateral horns, avoiding the midline'' but actually there are 3 different sites for stunning are applied (month 1,2,3)

line 115: you indicated that '' bleeding time, acceptance criterion (accepted or excluded) and any observations taken during each evaluation''  but such results not mentioned in the paragraph of  results.

line 197-199: table 1.  the results of phase I of study were collected together although it's from 3 different sites of stunning procedure (data from 3 months, data of  each  month were from different stunning sits), why you not separated the results (for weight, sex and breed) like in the figure 3?        

line 197-199: table 1. phase I, PPCB shot 2  were 68animals,  but in the text (lines 159-183) are 37 (month 1) + 26 (month 2) + 6 (month 3) = 69!  also in table 1, for phase I, firearm stunning were 61 animals, but in the text (lines 159-183) are 34(month 1) + 22 (month 2) + 4 (month 3) = 60! 

why 2 cases of firearm stunning from phase II of study not included in the table 2 for brain damage degree evaluation ?

lines 238-239 : in table 2 corrected the word ''dammage'' to ''damage''

line 396: bleeding time not mentioned in the paragraph of results, why?  

you do not discuss the reasons, why the differences in the results of stunning effectiveness of phase I/ month 3 (first  shot was achieved in 85.0%, the second shot was 9.0% and in 6.0%)  and phase II (the first shot was achieved in 95.1%; in a second shot was required 3.9%, and firearm 1.1%) although the stunning sites were the same for the both

Author Response

Response to Reviewer 3 Comments

The factor of animal’s age which is not considered in the study can be play important role on the stunning effectiveness

Response: Agree. However, due to the characteristics and dynamics of the slaughterhouse, it was difficult to follow up on the determination of age. Our study was focused on the stunning effectiveness based on the presence of unconsciousness signs of high discriminatory power (evaluated in the stunning box) regardless age of all buffaloes slaughtered during the trial months. The results obtained are novel and achieved a highly effective first-shot stunning in water buffaloes. However, we are very positive to continue researching on this topic, and further studies are going to consider including age as a variable of analysis.

Line 66: you indicated that the abattoir slaughters approximately 90 to 100 buffaloes are processed 66 monthly, but in the study at phase I/ month 1 and 2 of study 143 and 141 buffaloes were slaughtered, respectively.

Response 66: In section 2.2 we mention an approximate number of buffaloes processed (“approximately 90 to 100 buffaloes are processed monthly”), this is an estimate, because the number of buffaloes recieved for slaughter is variable.

Lines 80-88: in paragraph study design you indicated that during phase I of study one site of stunning was used '' in a frontal area, initially referenced by Glardon et al. (2018): the intersection of two imaginary lines that join the upper and lower edges of the contralateral horns, avoiding the midline'' but actually there are 3 different sites for stunning are applied (month 1,2,3)

Response 80-88: The study was initially designed taking Glardon et al (2018) as a reference but using a PPCB. During the evaluation of the postmortem heads of that first month, there was a slight narrowing that was observed in the skulls of the buffaloes at approximately 8 cm dorsal to the intersection of the imaginary lines connecting the horn with the contralateral eye. For that reason, from Month 2, that was our new entry point used. During Month 3, repeatability was given to that new entry point observed in month 2. Therefore, only 2 stunning sites were used.

Line 115: you indicated that '' bleeding time, acceptance criterion (accepted or excluded) and any observations taken during each evaluation'' but such results not mentioned in the paragraph of results.

Response 115: Agree. The text should be modified since the acceptance criteria referred to the selection of animals described in point 2.3, and the animals evaluated are those that met the requirements. Regarding the bleeding time, we are going to include it in the methodology (section 2.5), since it was a requirement of the slaughter process; however, this variable remained constant (less than 60 seconds) throughout the study. For this reason, we did not consider including it as a result.

Line 197-199: table 1.  the results of phase I of study were collected together although it's from 3 different sites of stunning procedure (data from 3 months, data of each month were from different stunning sits), why you not separated the results (for weight, sex and breed) like in the figure 3?       

Response 197-199: Only 2 stunning sites explained in response 80-88 were used. Phase I was considered to evaluate frontal stunning using a specific PPCB (Jarvis) that to date had not been considered in any other study in water buffalo. The second phase consisted of validating and repeating the newfound site. Table 1 presents the comparative approach of the validation phase (Phase II) concerning phase I.

Line 197-199: table 1. phase I, PPCB shot 2 were 68 animals, but in the text (lines 159-183) are 37 (month 1) + 26 (month 2) + 6 (month 3) = 69!  also in table 1, for phase I, firearm stunning were 61 animals, but in the text (lines 159-183) are 34(month 1) + 22 (month 2) + 4 (month 3) = 60!

Response 197-199: Agree, the information of the text is correct, we are going to modify the Table information.

Why 2 cases of firearm stunning from phase II of study not included in the table 2 for brain damage degree evaluation?

Response: The 2 cases of stunning with a firearm (secondary stunning method) were not included because the assessment of the degree of damage was carried out to demonstrate the damage generated by the PPCB as the primary method of stunning. The damage produced in brain tissue using a firearm already been described in other studies. Our approach was the degree of damage generated according to the effectiveness of the primary method.

Lines 238-239: in table 2 corrected the word ''dammage'' to ''damage''

Response 238-239: Agree. We are going to modify it.

Line 396: bleeding time not mentioned in the paragraph of results, why? 

Response 396: the exsanguination process was established to be performed in less than 60 seconds after the stun. It was a requirement of our study. According to response 115 (above) we are going to include it in the methodology (section 2.5).

You do not discuss the reasons, why the differences in the results of stunning effectiveness of phase I/ month 3 (first shot was achieved in 85.0%, the second shot was 9.0% and in 6.0%)  and phase II (the first shot was achieved in 95.1%; in a second shot was required 3.9%, and firearm 1.1%) although the stunning sites were the same for the both

Response: : It is important to clarify that Phase 2 is a consecutive step of Phase 1. Phase 2 uses the same anatomic location as used in Phase 1. During Phase 2 two major determinants were improved: the stunning personnel was in a learning curve process to adapt a newer site to improve the accuracy of the buffaloes stunning procedure. The increased values regarding effective stunning at the first shot obtained in Month 2 and specially in Month 3 (85%) lead us to the conclusion that this was an appropriate frontal approach.

After a planned preventive maintenance of the PPCB, and the change the compressed air maintenance unit, there was a considerable improvement in the stunning results. In addition, the stunning personnel were better trained to identify the new shot entry. Those situations, led as to define both Phases.

Phase 1 evaluated the feasibility of frontal stunning with Jarvis USSS-1 PPCB, and in Phase 2 validated the effectiveness and repeatability of adequate stunning in this proposed area.

Round 2

Reviewer 3 Report

Answers of the authors for my comments are acceptable 

Author Response

We are very thankful for your revision.